# Design and Optimization of GeSn Waveguide Photodetectors for 2-µm Band Silicon Photonics

**DOI:** 10.3390/s22113978

**Published:** 2022-05-24

**Authors:** Soumava Ghosh, Radhika Bansal, Greg Sun, Richard A. Soref, Hung-Hsiang Cheng, Guo-En Chang

**Affiliations:** 1Institute of Radio Physics and Electronics, University of Calcutta, Kolkata 700009, India; ghoshsoumava2@gmail.com; 2Department of Mechanical Engineering, and Advanced Institute of Manufacturing with High–Tech Innovations (AIM–HI), National Chung Cheng University, Chiayi County 62102, Taiwan; d09420005@ccu.edu.tw; 3Department of Engineering, University of Massachusetts-Boston, Boston, MA 02125, USA; greg.sun@umb.edu (G.S.); richard.soref@umb.edu (R.A.S.); 4Center for Condensed Matter Sciences and Graduate Institute of Electronics Engineering, National Taiwan University, Taipei 10617, Taiwan; hhcheng@ntu.edu.tw

**Keywords:** waveguide photodetector, saturation velocity, *R*_0_*A* parameter, responsivity, bandwidth, detectivity, silicon photonics

## Abstract

Silicon photonics is emerging as a competitive platform for electronic–photonic integrated circuits (EPICs) in the 2 µm wavelength band where GeSn photodetectors (PDs) have proven to be efficient PDs. In this paper, we present a comprehensive theoretical study of GeSn vertical *p–i–n* homojunction waveguide photodetectors (WGPDs) that have a strain-free and defect-free GeSn active layer for 2 µm Si-based EPICs. The use of a narrow-gap GeSn alloy as the active layer can fully cover entire the 2 µm wavelength band. The waveguide structure allows for decoupling the photon-absorbing path and the carrier collection path, thereby allowing for the simultaneous achievement of high-responsivity and high-bandwidth (BW) operation at the 2 µm wavelength band. We present the theoretical models to calculate the carrier saturation velocities, optical absorption coefficient, responsivity, 3-dB bandwidth, zero-bias resistance, and detectivity, and optimize this device structure to achieve highest performance at the 2 µm wavelength band. The results indicate that the performance of the GeSn WGPD has a strong dependence on the Sn composition and geometric parameters. The optimally designed GeSn WGPD with a 10% Sn concentration can give responsivity of 1.55 A/W, detectivity of 6.12 × 10^10^ cmHz^½^W^−1^ at 2 µm wavelength, and ~97 GHz BW. Therefore, this optimally designed GeSn WGPD is a potential candidate for silicon photonic EPICs offering high-speed optical communications.

## 1. Introduction

Fiber optic communication systems offer a strong competition to their electronic counterparts due to their low interference, low attenuation (~0.2 dB/km), and high operational speed. Although the 1.31 µm and 1.55 µm wavelength bands are very popular in the telecommunication industry, the exponential growth of internet traffic has driven their data transport capacity to the verge of its theoretical limit [1]. In addition, their bandwidths (BWs) can no longer be significantly increased using the existing technologies. Therefore, researchers are trying to develop new technologies to circumvent this bottleneck condition. A significant approach to overcome this aforesaid shortcoming is to use hollow-core photonic bandgap fibers (HC–PBGFs), which provide a low loss (~0.1 dB/km) with a high speed ~0.98 times of *c* (velocity of light in free space) in the 2 µm wavelength band, i.e., the wavelength range of 1.8–2.1 µm [2]. The photonic bandgap is effective to control and confine the propagation of electromagnetic waves irrespective of their polarization [3,4,5]. A notable application of the photonic bandgap is photonic waveguide. Moreover, high-BW thulium-doped fiber amplifier (TDFA) light source that can emit light in the range of 1.8–2 µm is suitable for the 2 µm wavelength band [6]. Those distinctive advantages have made the 2 µm wavelength band a supplement to the 1.31 and 1.55 µm wavelength bands for next-generation high-speed fiber optic communication. Apart from telecommunication applications, this 2 µm band can also be used in sensing and biomedical applications [7].

Photodetectors (PDs) are an essential photonic device to convert optical signals to electrical signals [8]. For 2 µm wavelength applications, although market-dominated III–V-based short–wave infrared (SWIR) PDs show a promising performance in the 2 µm wavelength [9,10], their incompatibility with Si–based complementary metal-oxide-semiconductor (CMOS) processing technology has made them expensive and less possible for large-scale integration. On the other hand, over the last few decades, group–IV–based SWIR PDs have attracted attention due to their compatibility with standard Si-based CMOS processing technology and their monolithically integrability on the same Si chip [11,12] to realize electronic–photonic integrated circuits (EPICs). However, Si- and Ge-based SWIR PDs are not suitable for 2 µm due to their short cutoff wavelengths of 1.1 µm and 1.5 µm, respectively [11,12,13].

There is a dramatic improvement of this situation in the last two decades, with the successful growth of a new group–IV material system, Ge_1−x_Sn_x_ alloys, using a suitable buffer layer on Si substrates-either by molecular beam epitaxy (MBE) [14,15] or by chemical vapor deposition (CVD) techniques [16]. Adding Sn into Ge can significantly reduce the direct bandgap, thereby redshifting the direct bandgap and thus enabling efficient photodetection in the infrared region. In addition, the standard Si-based CMOS compatibility, bandgap tunability due to Sn incorporation, direct bandgap for x ≥ 0.06 [17], large absorption coefficient in the infrared region [18] and high carrier mobility [19] make Ge_1−x_Sn_x_ alloys attractive as active materials for different types of PDs [18,20,21,22,23,24,25,26,27,28,29]. A variety of normal–incidence Ge_1−x_Sn_x_–based *p–i–n* PDs have been demonstrated by a few groups, showing promising performance in the SWIR region, and even up to 3.7 µm [30,31,32,33,34,35,36]. A recent theoretical study has predicted that normal-incidence GeSn PDs’ performance can be comparable to, and even better than, conventional SWIR and MIR PDs [17]. However, for EPICs, planar waveguide PDs (WGPDs) are preferred for more convenient integration with other photonic devices. In addition, planar WGPDs have many unique advantages over normal-incidence PDs. First, the decoupling of the photon absorption path and the photogenerated carrier collection path in WGPDs removes the trade-off between spectral responsivity and 3-dB bandwidth (BW). Second, the long photon-absorbing path can sufficiently absorb the incident photons, thus significantly increasing the optical responsivity. This salient feature makes the WGPDs very effective for modern high-speed communication applications. To date, only few attempts have been made to fabricate WGPDs based on bulk GeSn alloys and multiple-quantum-well (MQW) structures [37,38,39,40]. Among those studies, researchers have primarily focused on the measurement of responsivity, dark and photocurrent, and 3-dB BW of the WGPDs [37,38,39,40], showing a performance superior to that of normal-incidence GeSn PDs. However, the performance of GeSn PDs is also highly dependent on the device geometry. It is therefore important to also optimize their geometry to predict achievable performance for practical applications at the 2 µm wavelength band.

In this paper, we propose and analyze a novel GeSn homojunction vertical *p–i–n* WGPD on Si substrate to obtain simultaneously high–BW and high responsivity at the 2 µm wavelength band currently considered as a promising communications window. First, we show our proposed design of GeSn WGPDs. Then, we present analytical models to calculate the saturation velocity of carriers in GeSn, the 3-dB BW, optical responsivity and the *R*_0_*A* parameters of the proposed GeSn WGPDs. Following that, we analyze the dependence of Sn content and structural parameters on the device performance and optimize the structural parameters of the proposed GeSn WGPDs to simultaneously achieve high-BW and high responsivity at 2 µm. Finally, we evaluate the detectivity of the optimally designed GeSn WGPDs and compare them with conventional SWIR and MIR PDs. These results provide useful guidelines for design high-performance GeSn WGPDs for silicon photonics operating at 2 µm wavelength band.

The rest of the paper is organized as follows: the structural architecture of the proposed WGPD is shown in Section 2; the theoretical models are summarized in Section 3; the performance analysis of the GeSn WGPD at 2 µm is discussed in Section 4; followed by the conclusion in Section 5.

## 2. Design of GeSn WGPDs

### 2.1. Design of GeSn p–i–n Waveguide Photodetectors

The structure we propose for the GeSn vertical *p–i–n* homojunction WGPD is depicted in Figure 1. The intrinsic Ge_1−x_Sn_x_ absorption region is sandwiched between two heavily doped *p*-and *n*-type Ge_1−x_Sn_x_ layers with a doping concentration of 1 × 10^19^ cm^−3^. The entire vertical *p–i–n* structure is grown on Si (001) or silicon-on-insulator (SOI) substrates via a compositionally step-graded, fully strain-relaxed Ge_1−x_Sn_x_ virtual substrate (VS) [17], whose thickness is set to 1000 nm, resulting in a lattice-matched strain-free GeSn *p–i–n* homojunction structure. In addition, to predict the highest performance, the GeSn *p–i–n* structure is assumed to be defect-free [17] and the waveguide coupling efficiency to be 100%. The thickness of the *p*–type Ge_1−x_Sn_x_, *i*–Ge_1−x_Sn_x_ active layer and *n*–Ge_1−x_Sn_x_ layers were *t_p_*, *t_i_* and *t_n_*, respectively. Here, we set *t_p_* = 500 nm and *t_n_* = 100 nm. The WGPD has a ridge structure down to the top surface of *p*–Ge_1−x_Sn_x_ and is covered by a SiO_2_ layer as the electrical isolator for the Ge_1−x_Sn_x_ homojunction and as the cladding layer for the waveguide. The width of the ridge structure (*w*) wasset to 1 µm for single-mode operation, while the length of photon absorption region *L* and the thickness of the photogenerated carriers’ collection path *t_i_* were optimized to achieve the highest performance. Under illumination, the electron hole pairs (EHPs) generated in the Ge_1−x_Sn_x_ region are swept to the *n*–Ge_1−x_Sn_x_ and *p*–Ge_1−x_Sn_x_ regions due to the built-in electric field, thereby producing photocurrents.

### 2.2. PD to Waveguide Coupling

Several assumptions were made in this paper regarding the PD that is detailed here: (1) the PD, or a group of PDs, is part of a waveguided photonic-integrated circuit (PIC) that is fabricated on a wafer substrate; (2) that wafer is most likely a SOI wafer, although it could alternatively be bulk silicon; (3) the PIC and typically some transistor–electronic integrated circuits are co-integrated on the SOI wafer by manufacturing in a Si CMOS foundry. Regarding the efficient optical coupling between the PD and a low-loss strip waveguide, there are several practical scenarios that are envisioned here. These feasible approaches are: (1) the undoped strip waveguide is comprised either of Si or Ge, but Si is preferred; (2) an SOI strip waveguide is end–fire coupled into the *i*–GeSn PD core without an air gap between the Si core and PD core because the selective area growth of the PD within an SOI “trench” is used to construct the PD; (3) or alternatively, the PD is built upon the top surface of the SOI strip in order to use the “evanescent–wave” side coupling of light along the long axis of the Si strip.

Going into more detail, we are proposing two practical approaches that are illustrated in Figure 1b,c, showing a cross–sectional side view along the direction of light propagation. Generally, light enters from the left and is mostly absorbed by the PD and then converted to photocurrents. The first scenario, Figure 1b, shows that buried SiO_2_ is locally removed over a small area. Oxide is etched away, down to bare Si, and in this trench, the GeSn virtual substrate is grown, which then is the foundation for the *p–i–n* PD. This is the in–line coupling or coaxial case in which the Si guided mode axis aligns exactly with the *i*–GeSn center line. The second scenario, Figure 1c, shows that the SOI top layer is not disturbed (except for local strip etching) and that the GeSn VS is grown first on that Si layer, after which the *p–i–n* PD is deposited upon the VS. From an optical point of view, the incoming light from the first Si strip is upwardly coupled into the GeSn regions because the GeSn constitutes a “leaky top cladding” for the Si since the GeSn has a refractive index (RI) higher than that of Si. This evanescent longitudinal and side coupling into GeSn gives the needed optical absorption.

## 3. Theoretical Models

In this section, we describe the theoretical models used to analyze the performance of the proposed GeSn WGPDs, including the carrier saturation velocities in GeSn material, the absorption coefficient, 3-dB BW, optical responsivity, *R*_0_*A* parameters, and detectivity. Due to the limited availability of experimental data for GeSn alloys, the material parameters of GeSn alloys used in this study were obtained from a linear interpolation between these of Ge and α–Sn [41]. With the availability of more experimental data, it is expected that our calculations can predict the device’s performance more accurately.

### 3.1. Absorption Coefficient

In Ge_1−x_Sn_x_, the photon absorption takes place due to (1) the direct bandgap interband transition from the valance band (VB) to the Γ–valley conduction band (CB), and (2) the indirect bandgap interband transition from the VB to the *L*-valley CB. The direct–band optical absorption coefficient (*α_dir_*) of Ge_1−x_Sn_x_ alloys can be calculated using the Fermi’s golden rule by taking into account the nonparabolicity effect and Lorentzian lineshape function as [17,18],
(1)$$\alpha_{dir}\left(\hbar \omega \right)=\frac{\pi \hbar e^{2}}{n_{r}c\varepsilon_{0}m_{0}^{2}\hbar \omega }\sum_{m}\int \frac{2dk}{{\left(2\pi \right)}^{3}}{\left|\hat{e}.p_{CV}\right|}^{2}\times \frac{\gamma /2\pi }{{\left[E_{C\Gamma }\left(k\right)-E_{m}\left(k\right)-\hbar \omega \right]}^{2}+{\left(\gamma /2\right)}^{2}}\;$$
where *n**_r_* represents the RI of the active medium, *c* is the velocity of light in vacuum, *ɛ*_0_ is the free space permittivity, *m*_0_ is the electron’s rest mass, *ω* is the angular frequency of incident light, *e* is the electronic charge, *ħ* is the reduced Planck’s constant, ${\left|\hat{e}\cdot p_{CV}\right|}^{2}=m_{0}E_{P}/6$ indicates the momentum matrix, *E_P_* denotes the optical energy, *γ* is the full–width–at–half–maximum (FWHM) of the Lorentzian lineshape function. *E_C_*_Γ_(**k**) and *E_m_*(**k**) represent the electron and hole energy in the Γ–valley CB and VB, respectively, where **k** is the wavevector, and are calculated using a multi–band k·p method [18,41,42]. The summations over *m* indicate all interband transitions from the VB, including the heavy hole (HH) and light hole (LH) bands, to the Γ–valley CB. On the other hand, the indirect band optical absorption coefficient (*α_indir_*) can be calculated using the following empirical expression [43],
(2)$$\alpha_{\mathrm{indir}}=A{\left(\hbar \omega -E_{g}^{L}+E_{ap}\right)}^{2}+A{\left(\hbar \omega -E_{g}^{L}-E_{ap}\right)}^{2}\;$$
where the first (second) term denotes the acoustic phonon absorption (emission) mechanism thatoccurs for $\hbar \omega >E_{g}^{L}-E_{ap}\left(\hbar \omega >E_{g}^{L}+E_{ap}\right)$, with $E_{g}^{L}$ denoting the indirect bandgap energy and with *E_ap_* being the energy of acoustic phonon. Due to the unavailability of GeSn related experimental data and the similarity of the band structures of GeSn and pure Ge, in this analysis the values for GeSn are approximated by these of pure Ge (*A* = 2717 cm^−1^ [17,44] and *E_ap_* = 27.7 meV at room temperature [45]). Thus, the total optical absorption coefficient (*α*) can be expressed as,
(3)$$\alpha =\alpha_{\mathrm{dir}}+\alpha_{\mathrm{indir}}\;$$

The penetration depth (*d*) can then be calculated by
(4)$$d=\frac{1}{\alpha }\;$$

### 3.2. Saturation Velocity and Bandwidth

The BW of a PD is dominated by two mechanisms, transit–time delay and RC delay. The transit–time–delay–limited BW (*f*_T_) can be calculated using [29]
(5)$$f_{T}=0.45\frac{v_{S}}{t_{i}}\;$$
where *t_i_* represents the thickness of the intrinsic Ge_1−x_Sn_x_ active region and *v_S_* is the carrier saturation velocity. Previously, the carrier saturation velocity was usually approximated to that of Ge due the lack of experimental data of GeSn. However, it is well known that the carrier mobilities of GeSn significantly increase with increasing Sn content [17,19]. As a result, the carrier saturation velocity of GeSn should vary significantly compared to pure Ge. Therefore, in this paper, we theoretically calculated the carriers’ saturation velocities of GeSn alloys. The carrier saturation velocity can be calculated by [46]
(6)$$v_{S}=\sqrt{\frac{\Delta E}{m^{*}\sigma_{s}N_{s}L_{me}}}\;$$
where $\Delta E=hc_{s}/a_{\mathrm{GeSn}}$, *h* is the Planck’s constant, *c_s_* is the velocity of sound in the GeSn material that can be obtained by $c_{s}=\sqrt{\mu /\rho }$ with *µ* being the shear modulus and *ρ* being the density, *a*_GeSn_ is the lattice constant of GeSn, *m** is the conductivity effective mass that is taken from the calculation results using the 30-band full-zone k·p method [47], *N_s_*$\left(=8/a_{\mathrm{GeSn}}^{3}\right)$ is the atomic density of GeSn alloys, $L_{me}=N_{s}^{-1/3}$ is the mean free path [46], and *σ_s_* is the capture cross–sectional area of the atom. Due to the lack of a value of capture cross–sectional area of GeSn alloys, in this analysis, we approximated the capture cross–sectional area using that of the pure Ge atom, i.e., *σ_s_* = 1 × 10^−17^ cm^−2^ [46]. The lattice constant of bulk Ge_1−x_Sn_x_ (*a*_GeSn_) can be calculated by [48]
(7)$$a_{\mathrm{GeSn}}=\left(1-x\right)a_{\mathrm{Ge}}+xa_{\mathrm{Sn}}+x\left(1-x\right)\theta_{\mathrm{GeSn}}\;$$
where *a*_Ge_ = 5.6573 Å and *a*_Sn_ = 6.4892 Å are the bulk lattice constants of Ge and Sn, respectively, and *θ*_GeSn_ = 0.041 Å is the bowing parameter [48]. The RC–delay–limited BW (*f**_RC_*), on the other hand, can be calculated using [29]
(8)$$f_{RC}=\frac{1}{2\pi RC}$$
where *C* is the capacitance of the intrinsic GeSn region, which can be calculated using $C={\varepsilon A_{d}/t}_{i}$ with *ɛ* being the permittivity of the GeSn alloy; $A_{d}\left(=w\times L\right)$ is the cross–sectional area of the active region; and *R* denotes the load resistance. In this analysis, the standardized RF impedance of *R* = 50 Ω was considered [28,29]. Note that the parasitic capacitance of the device was not considered in this study. The parasitic capacitance of the device can increase the RC time delay, but it be minimized by optimizing the design of the electrodes. Using Equations (5) and (8), the total 3-dB BW can be calculated as [29]
(9)$$f_{3\mathrm{dB}}=\sqrt{\frac{1}{f_{T}^{-2}+f_{\mathrm{RC}}^{-2}}}$$

With the 3-dB bandwidth, the response time ($\tau_{r}$) can be calculated using
(10)$$\tau_{r}≅\frac{0.35}{f_{3\mathrm{dB}}}$$

### 3.3. Optical Responsivity

The responsivity (*R_λ_*) of WGPDs can be calculated using [49]
(11)$$R_{\lambda }=\kappa \frac{e\lambda \eta_{i}}{hc}\frac{\Gamma \alpha }{\Gamma \alpha +\alpha_{i}}\left(1-R_{s}\right)\left\{1-\exp\left[-\left(\Gamma \alpha +\alpha_{i}\right)L\right]\right\}$$
where *κ* is coupling efficiency, *λ* is the wavelength of the incident light wave, *η_i_* represents the internal quantum efficiency of the WGPD, Γ denotes the optical confinement factor (OCF) of the intrinsic Ge_1−x_Sn_x_ region, *α* is the total absorption coefficient of the active region given in Equation (3), *α_i_* indicates the internal absorption loss in the waveguide that does not contribute to photocurrents, and *R_s_* is the reflectivity of the WGPD. For maximum response, we assumed no coupling loss (*κ* = 1), perfect internal quantum efficiency (*η_i_* = 100%), and zero reflection (*R_s_* = 0), with the understanding that any less–than–ideal conditions can be accounted for proportionally.

### 3.4. Dark Current

In this defect–free GeSn WGPD, the minority carriers’ diffusion is the primary source of the dark current and the Shockley–Read–Hall (SRH) recombination process in the materials and at the GeSn sidewalls are not considered [50] as mentioned in Ref. [17]. Therefore, by considering the short-base diode approximation, the dark current density (*J*_dark_) can be calculated using [17]
(12)$$J_{\mathrm{dark}}=J_{\mathrm{diff}}^{\Gamma }+J_{\mathrm{diff}}^{L}+J_{\mathrm{diff}}^{h}=J_{0}\times \left[\exp\left(\frac{eV}{k_{B}T}\right)-1\right]$$
where *V* is the applied bias, *k*_B_ is the Boltzmann constant, *T* is the temperature and *J*_0_ indicates the reverse saturation current density, which can be expressed in terms of diffusion of electrons in Γ– and *L*–valley and holes as [17],
(13)$$J_{0}=e\frac{D_{n}^{\Gamma }}{t_{p}}n_{p0}^{\Gamma }+e\frac{D_{n}^{L}}{t_{p}}n_{p0}^{L}+e\frac{D_{p}}{t_{n}}p_{n0}$$
where $D_{p}$,$D_{n}^{\Gamma }$ and $D_{n}^{L}$ represent the diffusion coefficients of holes and electrons in the Γ– and *L*–valley, respectively; and $p_{n0}$, $n_{p0}^{\Gamma }$ and $n_{p0}^{L}$ denote the minority concentration of holes and electrons in the Γ– and *L*–valley, respectively.

### 3.5. R_0_A Product and Detectivity

The differential zero–bias resistance (*R*_0_) can be calculated by [51],
(14)$$R_{0}=\frac{k_{B}T}{eI_{0}}$$
where $I_{0}$ is the diode reverse current. From Equation (13), we can define *R*_0_*A* parameter in terms of the *J*_0_ as,
(15)$$R_{0}A=\frac{k_{B}T}{eJ_{0}}$$

The detectivity (*D^*^*) of the GeSn WGPD can be expressed as [26,42],
(16)$$D^{*}=\frac{R_{\lambda }\sqrt{A_{d}\Delta f}}{\sqrt{\langle i_{n}^{2}\rangle }}$$
where Δ*f* denotes the BW. Under dark condition, the thermal noise can be evaluated by [17,42],
(17)$$\langle i_{n}^{2}\rangle =\frac{4k_{B}T\Delta f}{R_{0}}$$

Hence, using Equations (14)–(16), the detectivity of the GeSn WGPD can be written as,
(18)$$D^{*}=\frac{R_{\lambda }}{2}\sqrt{\frac{R_{0}A}{k_{B}T}}\;$$

## 4. Results and Discussions

In this section, we first investigated the absorption spectra, penetration depth and the OCF of the GeSn active layer and then calculated the carriers’ saturation velocity of the holes and electrons in the Γ– and *L*–valleys to estimate the BW. After that, we optimized the different structural parameters of the proposed GeSn WGPD, including the thickness of the intrinsic region (*t_i_*), length of the photon absorbing layer (*L*), and Sn concentration (*x*) of the GeSn active region, to achieve the highest responsivity–bandwidth product (RBWP) at *λ* = 2 µm. Finally, we calculated the detectivity of the optimally designed GeSn WGPDs.

### 4.1. Absorption Coefficient

Figure 2a shows the calculated total optical absorption coefficient and penetration depth spectra with different Sn concentrations. It can be noted that, for a particular Sn concentration, the absorption coefficient decreases with increasing wavelength. As a result, the penetration depth also increases, suggesting that a longer device length is required for effective absorption in the WGPD. The increase inSn concentration reduces the direct and indirect bandgaps of the GeSn alloys [17]. Therefore, the absorption edge shows a redshift, thereby extending the photodetection range towards longer wavelengths and reducing the penetration depth. Figure 2b shows the calculated total absorption coefficient and penetration depth at *λ* = 2 µm with different Sn concentrations. The bandgap of the Ge_1−x_Sn_x_ alloy (x < 6%) is not small enough, leading to a very small absorption coefficient at *λ* = 2 µm. Thus, to obtain a high absorption (>10^3^ cm^−1^) at *λ* = 2 µm for efficient photodetection, the Sn concentration should be higher than 6%. On the other hand, for x < 6%, the penetration depth of >100 µm proves that the GeSn WGPD with a lower Sn concentration requires a longer device length for the sufficient absorption of the incident photons.

### 4.2. Optical Confinement Factor

To investigate the OCF of the GeSn WGPDs, we obtained various modes by using the finite element method (FEM), where the RIs of the materials we are taken from Refs. [27,44]. The simulated energy distributions of the quasi–transverse–electric fundamental (TE_00_) mode at *λ* = 2 µm for different Sn concentrations with *t_i_* = 1000 nm are illustrated in Figure 3 (for TE_00_ mode, the dominant fields are *E_x_* and *H_y_*). For *x* = 0%, there is no difference in RI between the *p–i–n* structure and the VS. As a result, the optical confinement is provided by the lower RI bulk Si substrate (*n* = 3.45) and the top SiO_2_ layer (*n* = 1.45). As a result, light cannot be properly confined in the *i*–Ge active layer, but leaks to the Ge VS and *p*–Ge layer, yielding a small OCF of 2.67% for the active layer. As the Sn concentration increases, the RI of the *i*–Ge_1−x_Sn_x_ active layer increases, thereby pushing up the optical field, and enhancing the optical confinement inside the *i*–Ge_1−x_Sn_x_ active layer. Figure 4 shows the calculated OCF for fundamental TE_00_ mode for the *i*–Ge_1−x_Sn_x_ active layer as a function of Sn concentration as well as the thickness of the active region at *λ*= 2 µm. The increase in the active layer thickness can significantly enhance the OCF because the *i*–Ge_1−x_Sn_x_ active layer occupies a larger portion of the waveguide. On the other hand, increasing the Sn composition can also increase the OCF for the GeSn active layer due to the increased RI difference between the GeSn active layer and the VS.

### 4.3. Carrier Saturation Velocity

In this paper, we theoretically analyzed the saturation velocity of the holes and electrons in the Γ– and *L*–valleys of GeSn using Equation (6). Figure 5 shows the calculated saturation velocities as a function of Sn concentration. The increase in Sn concentration reduces the effective masses of the carriers, and thus enhances the carrier mobilities [17]. As a result, the saturation velocities increase with the increase in Sn concentration. The analysis suggests that carriers take less time to transit through the active layer, thereby being beneficial for increasing the transit–time–delay–limited BW. For the saturation velocity of electrons, as the electrons in Γ–valley CB have a much smaller effective mass than that of the *L*–valley electrons [17,47], they have a much higher saturation velocity than these in the *L*–valley CB. On the other hand, the hole saturation velocity is much smaller than the electron saturation velocities due to the larger effective mass. Therefore, the holes take a longer time to transit through the Ge_1−x_Sn_x_ active region than the electrons. As a result, the holes’ movement entirely dominates the transit–time–delay–limited BW of the GeSn WGPDs.

### 4.4. Bandwidth

Next, we calculated the transit–time–delay BW, RC–delay BW and 3-dB BW for the Ge_1–x_Sn_x_ WGPDs using Equations (5), (8) and (9). For the transit–time–delay, we considered the hole time delay as it is significantly larger than the electron time delay as discussed in Section 4 (Section 4.3). Figure 6 shows the transit–time–limited BW, RC–limited BW and 3-dB BW as a function of the GeSn active region thickness with *x* = 6% and a device length of *L* = 50 µm. When the thickness of active region increases, the holes require longer time to transit through the entire region, leading to a decreased transit–time–delay–limited BW. On the other hand, the capacitance of the intrinsic Ge_1−x_Sn_x_ active layer reduces with increased GeSn active thickness, so the RC–delay–limited BW increases. Therefore, the total 3-dB BW first increases with the increase of intrinsic Ge_1−x_Sn_x_ thickness, reaches a peak value, and then decreases with the further increase of its thickness. The peak 3-dB BW of ~91 GHz can be achieved for 300 nm thick Ge_1−x_Sn_x_ active layer. Similar results were obtained for other Sn concentrations and other lengths.

### 4.5. Optimization of the GeSn WGPD at 2 µm

In this section, we optimized different structural parameters of our proposed GeSn WGPD to achieve a high–BW and high responsivity simultaneously at *λ* = 2 µm. First, we depict the responsivity and 3-dB BW as a function of thickness of the Ge_1−x_Sn_x_ active region in Figure 7a to obtain the optimum thickness of the Ge_1−x_Sn_x_ active region. The responsivity increases with increase of GeSn active layer thickness due to the increased OCF. For *t_i_* > 800 nm, the responsivity exhibits a saturation trend. On the other hand, the 3-dB BW first increases with the increased thickness of the GeSn active layer, reaches a peak value of 91 GHz at *t_i_* = 400 nm, and then decreases with the further increase in thickness. Between *t_i_* = 300–600 nm, the BW achieves more than 60 GHz with responsivity of *R_λ_* > 0.8 A/W. For optimal thickness, we calculated RBWP using Equations (9) and (10) and plotted it as a function of Ge_1−x_Sn_x_ active region thickness in Figure 7b. The RBWP shows a peak value at *t_i_* = 400 nm (marked by the red arrow), which is considered as the optimal thickness of the Ge_1−x_Sn_x_ active layer to achieve a high–BW and high responsivity simultaneously at *λ* = 2 µm. It is also noted that the OCF of the GeSn active can significantly impact the responsivity and RBWP. The OCF of the GeSn active layer can be enhanced by reducing the total thickness of the GeSn VS. In this way, the optical responsivity and RBWP of the devices can be further enhanced.

After obtaining the optimal thickness of the GeSn active layer, next we investigated the optimal device length (*L*_0_) to obtain the highest performance at *λ* = 2 µm. Figure 8a depicts the variation of the responsivity as well as the total 3-dB BW with the device length (*x* = 6%). It can be noted that, for 1 µm device length, our proposed GeSn WGPD can yield a high BW of ~98 GHz, but the corresponding responsivity is only ~0.036 A/W. Increase of device length increases the effective area and thereby the capacitance, so the RC–delay–limited BW is reduced as well as the total 3-dB BW. In addition, the responsivity increases with the increase of device length due to the longer photon–absorbing path. In order to obtain the optimal device length to achieve a high-BW and high responsivity simultaneously at *λ* = 2 µm, we therefore evaluated RBWP as a function of device length as shown in Figure 8b. The peak RBWP occurs at *L* = 70 µm (marked by the red arrow), corresponding to a BW of ~77 GHz and a responsivity of 1.275 A/W. Thus, *L* = 70 µm is considered as the optimal length of the device. For other Sn compositions, the optimal device length to achieve highest RBWP is also obtained using the same procedures. Figure 9a shows the variation of the optimum length of the GeSn WGPD with different Sn concentrations at *λ* = 2 µm. For pure Ge, to achieve the highest RBWP at 2 µm wavelength, a very long device length of *L*_0_ = 1130 µm is needed due to the negligible absorption coefficient. With the incorporation of Sn, the optimal length decreases with the Sn concentration *x* due to the enhanced absorption coefficient and reduced penetration depth of the GeSn active layer. For *x* > 6%, the optimal length decreases to only *L*_0_ < 70 µm. The small device length also indicates a small device footprint for high density EPICs. With the optimal length of the device, we then calculated the responsivity and 3-dB bandwidth of the optimized GeSn WGPDs, and the results are depicted in Figure 9b. For *x* < 6%, both the responsivity and 3-dB BW are small due to the small absorption coefficient. For the Sn composition ranging from 6% to 14%, the responsivity and 3-dB bandwidth can be significantly enhanced to achieve a high-performance photodetection at *λ* = 2 µm. In addition, the responsivity does not significantly change for 6% < *x* < 14%. Therefore, we considered *x* = 10% with *L*_0_ = 32 µm as the optimal length of the GeSn WGPD.

### 4.6. Responsivity

Based on the optimized geometrical structure (*t_i_* = 400 nm and *L*_0_ = 32 µm) of our proposed GeSn WGPD, we calculated the responsivity spectra with different Sn concentrations, and the results are shown in Figure 10. The increase of Sn concentration increases the responsivity and extends the photodetection capacity towards the longer wavelength. For *x* < 6%, the Ge_1−x_Sn_x_ alloy cannot effectively absorb the photons at *λ* = 2 µm due to its higher bandgap, leading to negligible responsivity. With *x* > 6%, the direct bandgap is significantly reduced, thereby enabling efficient photodetection at *λ* = 2 µm. For *x* > 8%, the photodetection range can fully cover the entire 2 µm wavelength band.

### 4.7. Dark Current and R_0_A Parameter

Using Equations (12) and (15), we calculated the dark current density (*J_dark_*) under zero–bias conditionand the *R*_0_*A* parameter of the GeSn WGPD as a function of Sn composition. Figure 11a illustrates the calculated *R*_0_*A* induced by different components and the total *R*_0_*A* parameter as well as the dark current density as a function of Sn concentration. The increase of Sn concentration reduces the bandgap, thereby increasing the intrinsic carrier concentration, then the diffusion currents [17]. Therefore, all the *R*_0_*A* components and the total *R*_0_*A* parameters decrease with the increase of Sn concentration. As *t_n_* << *t_p_*, the hole *R*_0_*A* components are dominant over the electron components. When the Sn concentration is lower than 6.6%, the Ge_1−x_Sn_x_ alloys are an indirect bandgap material [17], thus the injected minority electrons populate the *L*–valley more, rather than the Γ–valley. Therefore, the *R*_0_*A* for *L*–valley electrons is lower than that of Γ–valley electrons. However, when the Sn concentration exceeds 6.6%, the Γ–valley CB is lower than the *L*–valley CB, so more electrons can populate Γ–valley CB. As a result, the *R*_0_*A* parameter of the Γ–valley electrons reduces rapidly. In that condition, the contribution of Γ–valley electrons is dominant over those of *L*–valley electrons.

### 4.8. Detectivity

With the responsivity of the optimally designed GeSn WGPD and *R*_0_*A* at *T* = 300 K, we then further calculated the detectivity spectra for the GeSn WGPD with different Sn concentrations (Figure 12) compared with those of selected SWIR and MIR PDs. For a fixed Sn composition, the detectivity value decreases with increasing wavelength, and becomes negligible at the direct-gap absorption edge (the direct bandgap energy). The tunability of bandgap due to the incorporation of Sn helps to cover different spectral range according to the applications of interest. With an increase of Sn content, the cutoff wavelength of detectivity redshifts, but the magnitude of detectivity also decreases. In comparison with the commercially available uncooled (*T* = 300 K) SWIR and MIR PDs [17], it was found that, although InGaAs PDs have the highest detectivity, the cutoff wavelength is only 1.7 µm, which hinders the applications for 2 µm wavelength band detection. The GeSn PDs with *x* = 8% can achieve photodetection range fully covering the entire 2 µm wavelength band with a detectivity of >9.48 × 10^10^ cmHz^½^W^−1^. In addition, the GeSn PDs with *x* = 10% can have performance superior than E–InGaAs PDs, and comparable to PbS PDs. It is also noted that the detectivity of GeSn WGPDs is also much higher than that of PbSe and InSb PDs. Thus, we can conclude that our proposed GeSn WGPD can be a strong competitor of these commercial devices.

MIR and SWIR PDs are usually operated at cryogenic temperatures. If the operating temperature of the GeSn WGPDs is lowered, there are several impacts. First, the bandgap energy of GeSn becomes larger [52], so the direct-gap absorption edge will blueshift, resulting in a reduced photodetection range. Second, the intrinsic carrier density decreases, thereby significantly suppressing the dark current. As a result, the detectivity of the GeSn WGPDs is significantly enhanced.

### 4.9. Optimum Structures and Optimum Performances of the GeSn WGPDs

In this section, we present the optimum structure as well as optimum performances of the GeSn WGPDs for different Sn concentration (6–14%). This comparative study is shown in Table 1.

### 4.10. Comparative Study of Different GeSn PDs

Table 2 contains a comparative study of the responsivity obtained from our optimally designed GeSn WGPD with previously reported different GeSn PDs at 2 µm. The proposed GeSn WGPD can achieve a better performance than previously reported PDs.

## 5. Conclusions

In conclusion, we designed and optimized a new type of homojunction waveguide–coupled GeSn PD to achieve an optimum performance at 2 µm wavelength in the Si–or SOI–wafer EPIC platform. First, we theoretically calculate the carrier saturation velocities of the GeSn alloy. The incorporation of Sn reduces the effective mass of the carriers and thus the saturation velocity increases above that of pure Ge, suggesting that the GeSn–based devices have a higher operational speed than the pure–Ge-based devices. By carefully designing the length of the photodetection layer, the thickness of the photo–generated carrier collection path and Sn concentration, our proposed GeSn WGPD with 10% Sn concentration can give a responsivity of 1.549 A/W with ~97 GHz BW and detectivity of 6.12 × 10^10^ cmHz^½^W^−1^ at room temperature. The compatibility with standard Si–based CMOS technology, monolithical integrability on the same Si chip, high operational speed as well as high responsivity and detectivity make this proposed device an effective candidate for state–of–the–art silicon photonics for high–speed communication applications at the 2 µm wavelength band.

## Figures and Tables

**Figure 1 sensors-22-03978-f001:**
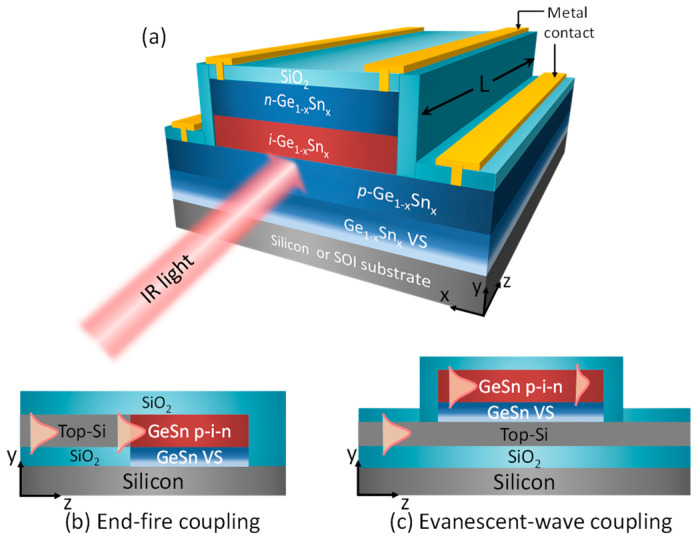
(**a**) Schematic diagram of our proposed homojunctionGeSn vertical *p–i–n* WGPD on Si or SOI substrate. Schematics of (**b**,**c**) coupling for the proposed GeSn WGPDs.

**Figure 2 sensors-22-03978-f002:**
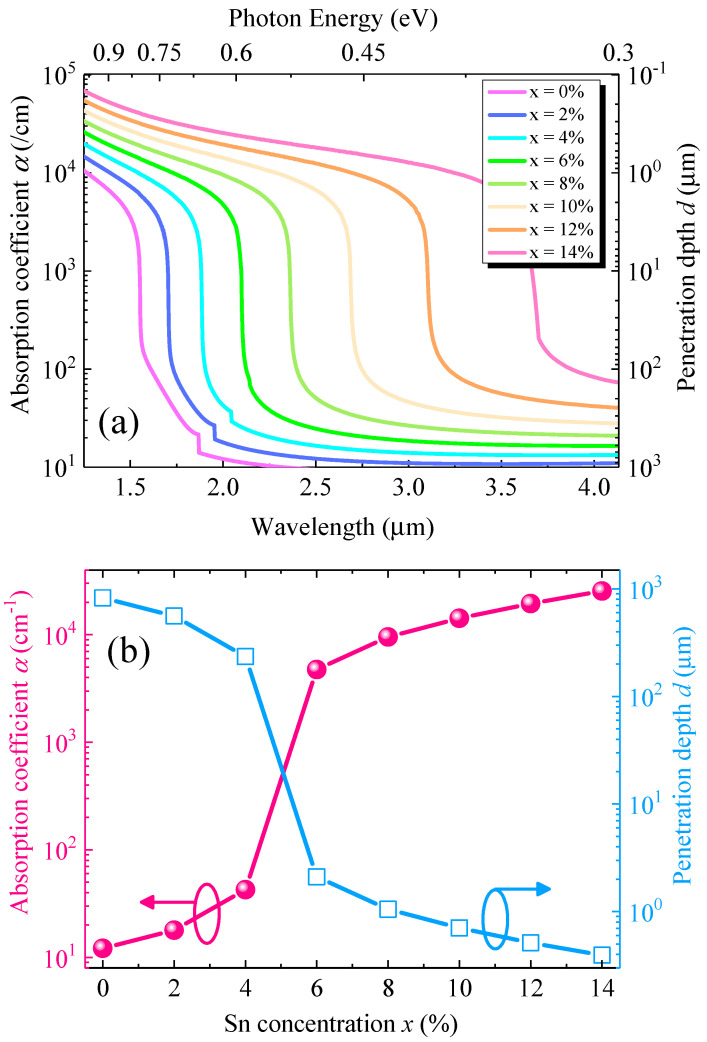
(**a**) Calculated total absorption coefficient and penetration depth spectra of GeSn WGPD for different Sn concentrations. (**b**) Calculated total absorption coefficient and penetration depth with various Sn concentrations at *λ* = 2 µm.

**Figure 3 sensors-22-03978-f003:**
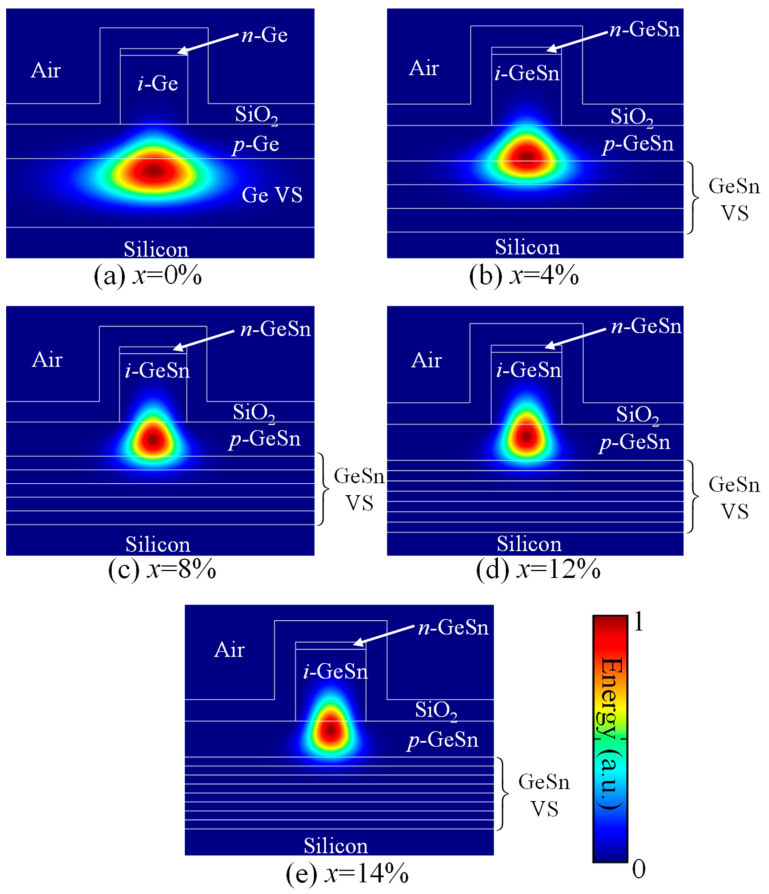
Simulated energy distribution for the quasi-transverse-electric fundamental mode of the GeSn WGPD for different Sn concentrations at *λ* = 2 µm.

**Figure 4 sensors-22-03978-f004:**
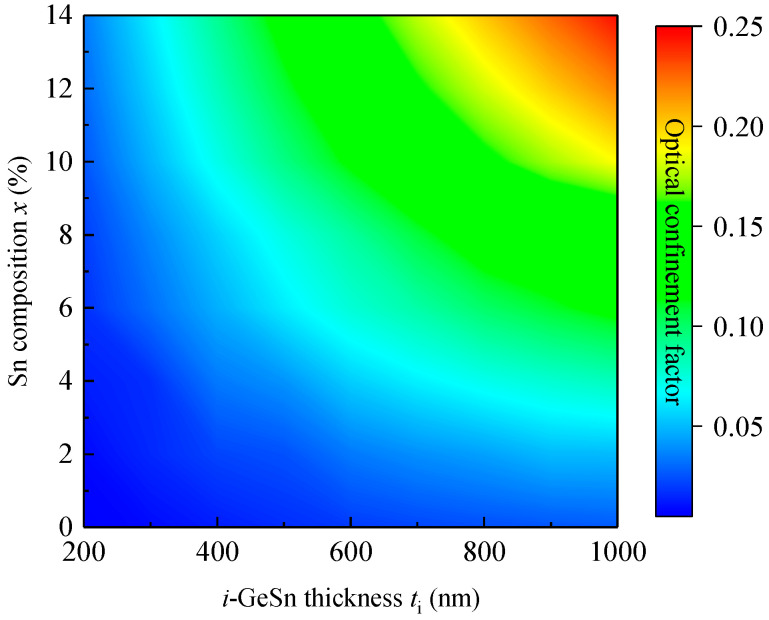
Simulated optical confinement factor for the quasi-transverse-electric fundamental mode for the GeSn active layer as a function of thickness and Sn concentration of the GeSn WGPD at *λ* = 2 µm.

**Figure 5 sensors-22-03978-f005:**
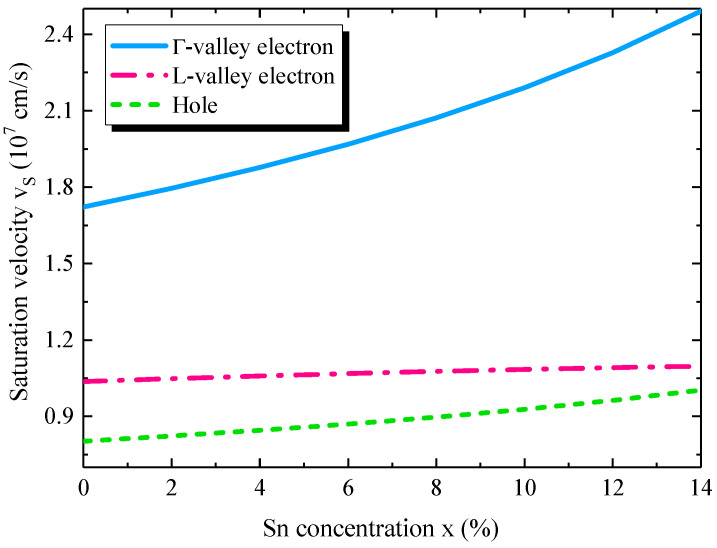
Calculated carrier saturation velocity as a function of Sn concentration inside the active region.

**Figure 6 sensors-22-03978-f006:**
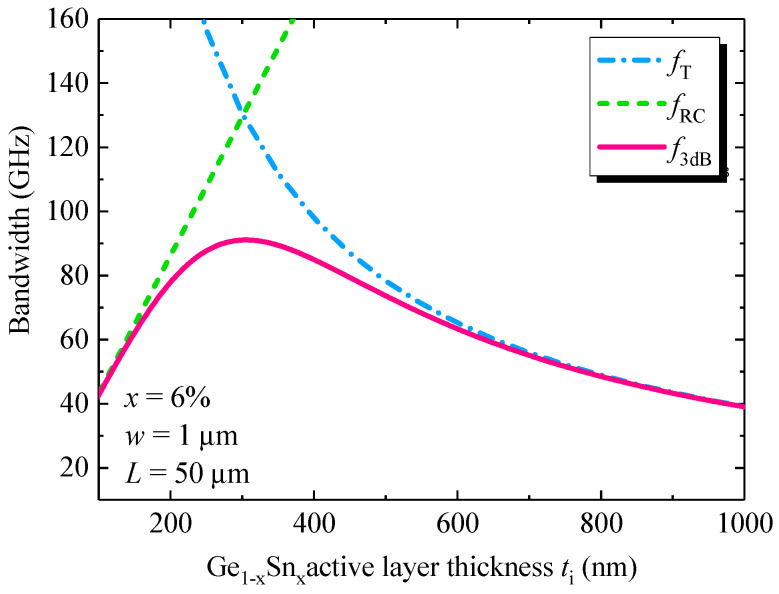
Calculated transit-time-delay-limited BW, RC-delay-limited BW and 3-dB BW as a function of Ge_1−x_Sn_x_ active layer thickness.

**Figure 7 sensors-22-03978-f007:**
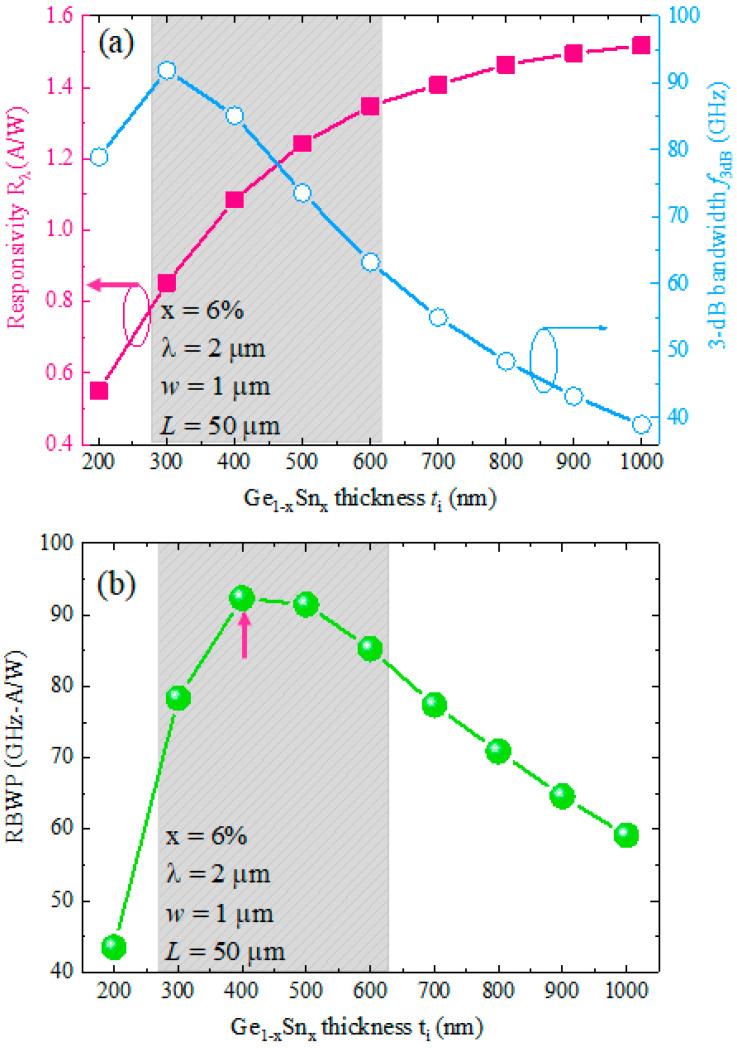
(**a**) Calculated responsivity and 3-dB bandwidth as a function of Ge_1−x_Sn_x_ active layer thickness at *λ* = 2 µm. (**b**) Calculated responsivity–bandwidth product (RBWP) against Ge_1−x_Sn_x_ thickness at *λ* = 2 µm.

**Figure 8 sensors-22-03978-f008:**
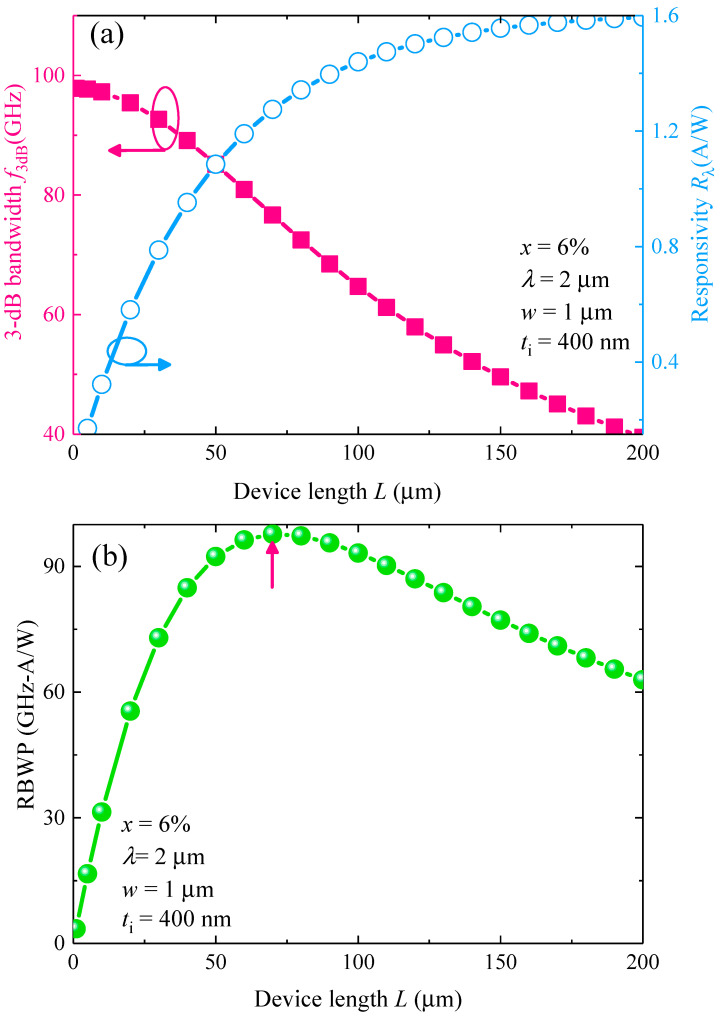
(**a**) Calculated 3-dB bandiwdth and responsivity as a function of device length at *λ* = 2 µm. (**b**) Calculated responsivity-bandwidth product (RBWP) as a function of device length at *λ* = 2 µm. The red arrow indicates the maximum RBWP.

**Figure 9 sensors-22-03978-f009:**
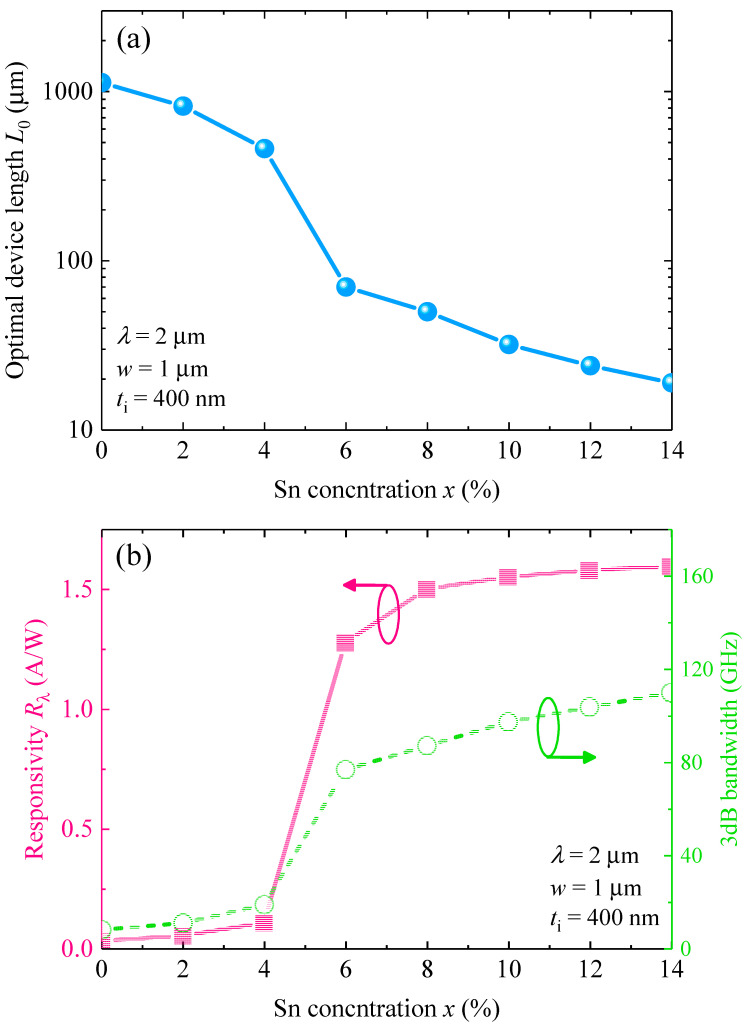
(**a**) Variation of optimum length of the GeSn waveguide photodetectors with various Sn concentrations at *λ* = 2 µm. (**b**) Variation of 3-dB bandwidth and responsivity against Sn concentration for optimized GeSn waveguide photodetectors.

**Figure 10 sensors-22-03978-f010:**
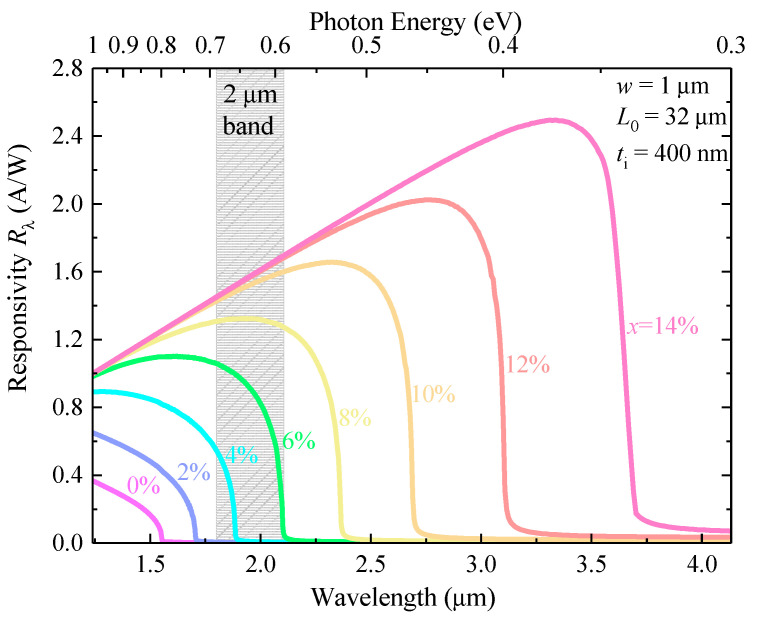
Calculated responsivity spectra of GeSn waveguide photodetectors for different Sn concentrations.

**Figure 11 sensors-22-03978-f011:**
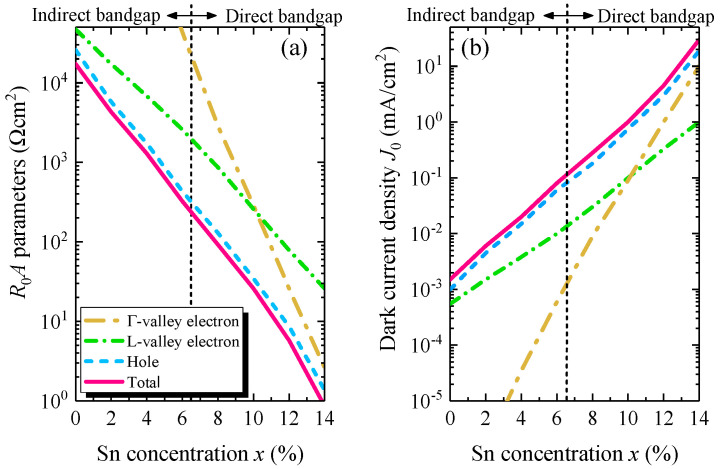
(**a**) Variation of different components and total *R*_0_*A* parameters and (**b**) dark current densities of GeSn WGPD with Sn concentration.

**Figure 12 sensors-22-03978-f012:**
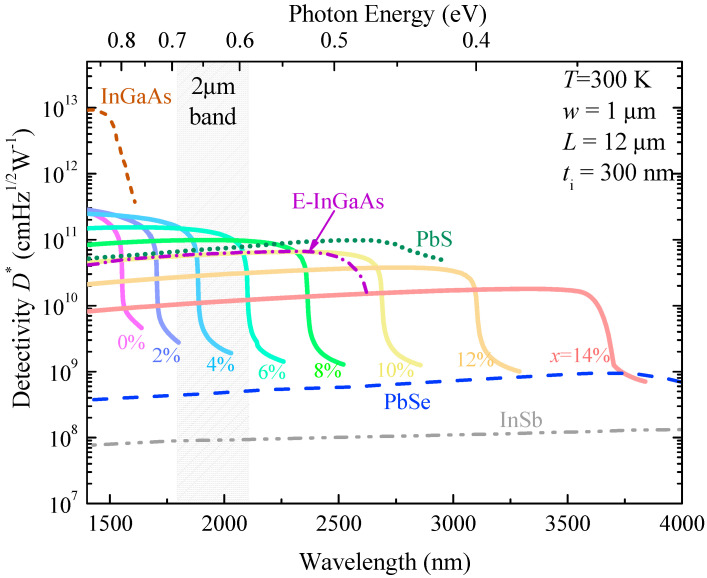
Calculated detectivity spectra of GeSn waveguide photodetectors with different Sn concentrations compared with selected SWIR and MIR PDs.

**Table 1 sensors-22-03978-t001:** Comparative study of the optimally designed GeSn WGPD for different Sn concentration at 2 µm wavelength.

Sn Content *x* (%)	Optimum Length *L*_0_ (µm)	Optimum Performance			
		Responsivity *R_λ_* (A/W)	Bandwidth *f*_3dB_ (GHz)	Response Time τ_r_ (ps)	Detectivity *D*^*^ (cmHz^½^W^−1^)
6	70	1.275	77	4.54	1.78 × 10^11^
8	50	1.499	87	4.02	1.12 × 10^11^
10	32	1.549	97	3.60	6.29 × 10^10^
12	24	1.578	103	3.39	2.94 × 10^10^
14	19	1.594	110	3.18	1.15 × 10^10^

**Table 2 sensors-22-03978-t002:** Comparative study of the optimally designed GeSn WGPD with previously reported GeSn PDs at *T* = 300 K.

Sn Concentration (%)	Substrate	Active Layer Thickness (nm)	Responsivity (A/W)	Reference
4.7	SOI	390	0.065	[53]
10	SOI	370	1.02	[28]
10	Si	270	1.51	[29]
4.3	Si	400	0.06	[40]
10	Si	3000	1.59	[17]
4.2	SOI	540	0.0022	[54]
10	Si	400	1.55	This work

## Data Availability

The data presented in this study are available upon request from the corresponding author. The data are not publicly available due to commercial privacy policy.

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
