# Peer review of "Design and Optimization of GeSn Waveguide Photodetectors for 2-µm Band Silicon Photonics"

_sensors, 2022, doi:10.3390/s22113978_

Round 1
Reviewer 1 Report
Authors showed comprehensive research of GeSn WGPDs with 3‐dB bandwidth, zero‐bias resistance, and detectivity. Literature background is enough to understand the previous research about GeSN WGPD. Theoretical analysis for bandwidth and zero‐bias resistance, and detectivity is very good with some concrete analysis. There are no English grammar issues. It is hard to find something wrong. Thus, the manuscript could be minor before publication.
*. In Eq. (9), whtat are fT and fRC ?
*. Figures 2 labels seems to be small.
*. Authors had better provide the literature for the sentence (Photodetectors (PDs) are an essential photonic device to convert optical signals~) with the reference (https://iopscience.iop.org/article/10.1088/1361-6560/ab6579/meta).
*. Line sizes in entire manuscript seems to be wrong. Please check MDPI format.
*. In Line 324, what is VS?
*. Data availability section ?
*. In Figure 6, fRC keep increasing. Is there any decreasing point ?
*. In Figure 7, responsivity keep increasing. Is there any saturation point ? As shown in Figure 10, there are saturation points.
Reviewer 2 Report
Please see the attached file.
